

# A process-based diagnosis of catchment coevolution in volcanic landscapes: synthesis of Newtonian and Darwinian approaches

Takeo Yoshida[1,2] and Peter A. Troch[1]

[1]University of Arizona, Tucson, AZ, USA
[2]National Agricultural and Food Research Organization, Tsukuba, Ibaraki, Japan

*Correspondence to:* Takeo Yoshida (takeoys@affrc.go.jp)

**Abstract.** Catchment coevolution is a framework that seeks to find a rigorous connection between landscape evolution and the emergent hydrological responses, and formulate hypotheses about how such evolution affects their hydrological response. Empirical studies previously conducted by the authors in volcanic catchments in Japan have revealed that the hydrological signatures baseflow index and slope of the flow duration curve (*SFDC*) decline with the age of the catchment bedrock; however, the possible influence of external climate forcing on these signatures could not be eliminated, and the causality behind the empirical relationships could not be identified. To test the robustness of the relationship and attempt to understand why such simple and predictable relations have emerged across a climate gradient, we used a process-based hydrological model that was independently calibrated for eight catchments underlain by volcanic rock of different ages and conducted a numerical experiment to decouple the effects of internal and external properties of the catchments. The eight calibrated catchment models were independently forced by the eight different sets of climate properties corresponding to the prevailing climates of the eight catchments to investigate how the simulated relationship between *SFDC* and catchment age deviates from the empirically derived relationship on both per catchment (each catchment forced by eight climates) and per climate (each climate filtered by eight catchments) basis. We found that the mean of the residuals of the observed versus predicted *SFDC* from each catchment, exhibited a significant positive correlation with catchment age, providing numerical evidence of catchment coevolution to support that of the empirical study. We further investigated the causality of this relationship and found from several simulated time scales and model fluxes that younger catchments on average (1) require longer for the transmission zone storage to fill and empty, (2) take longer to release water from deep aquifers, and (3) have greater recharge to deep aquifers. These findings corroborate the hypothesis of coevolution of volcanic catchments, which is that in younger catchments more water percolates to the subsurface storage and the deeper hydrologically active systems release water at a slower rate. The analysis also revealed that the external climate characteristics interact with the catchment internal properties in forming the catchment hydrological responses. The mean throughfall rate, which is controlled by rainfall intensity and should



be independent of catchment internal properties, significantly declined with catchment age in our
data set. The mean transpiration rate was the only candidate for a causal link between climate and
the hydrological signatures.

## 1  Introduction


The hydrological behavior of catchments is highly unpredictable because of the interactions of a
myriad of processes exerted across many different spatial and temporal scales (Sivapalan, 2003;
Wagener et al., 2007). Although the catchments are deemed as complex systems at a glance, the
present day landscapes have coevolved as self-organized systems through the interactions of cli-

mate, vegetation, bedrock geology and soils. The results of coevolution that can be captured in the
spatial patterns of landscape features including vegetation, geometry of stream networks, soil profiles
control the hydrological response of the catchments. Recognizing this interconections, hydrologists
have recently proposed the idea of catchment coevolution (Sivapalan et al., 2012; Troch et al., 2013;
Harman and Troch, 2014; Troch et al., 2015). In the framework of the catchment coevolution, one

seeks to find an empirical relationship between landscape features and the hydrological response,
and to decipher the causality of the relationship from the historical development of catchments.

However, because the initial conditions and processes involved during the landscape evolution are
unknown, a detailed understanding of how catchments have evolved to its present state is intimidat-
ing. Instead of classic Newtonian approach that studies one catchment in detail, an alternative ap-

proach, called Darwinian approach, was proposed (Wagener et al., 2013; Harman and Troch, 2014).
Darwinian approach in catchment hydrology first seeks hydrological variation to be explained in
a large sample of catchments and conceives a hypothesis that accounts for the patterns from the
historical perspective. Then, a set of consequences are derived that also must hold if the hypothe-
sis is true, and finally one can critically test the hypotheses by asking if these consequences hold

(Harman and Troch, 2014). Of course, due to the complexity of the landscape evolution, causality
may not always be as clear as in the Newtonian approach. One way to test the hypothesis and the
robustness of the emergent patterns is to use process-based hydrological models (Troch et al., 2013).
The advantage of using process-based hydrological models is that the causality behind responses
can be inferred from an examination of the hydrological processes represented in the model. Also,

numerical analysis enables to decipher which factors explain the observed hydrological behavior of
catchments.

Empirical studies previously conducted by the authors in volcanic catchments in Japan have re-
vealed that the hydrological signatures (baseflow index, *BFI*, and slope of the flow duration curve,
*SFDC*) significantly correlated with the age of the catchment bedrock (Yoshida and Troch, 2016),

which corroborated the findings of Jefferson et al. (2010) in the Oregon Cascades. Yoshida and Troch
(2016) analysed geomorphological features of the catchments as well as hydrological signatures and



hypothesized that the younger catchments exhibit groundwater-dominant response because most of the rainfall infiltrates and percolates into the permeable bedrock, which typically contains many cracks and fissures (Lohse and Dietrich, 2005). Older catchments on the other hand exhibit flashier

hydrographs because less permeable clay layers at shallow depth, formed by chemical weathering and mineral precipitation, thus impeding vertical recharge that cause shallow subsurface flow (Jefferson et al., 2010; Yoshida and Troch, 2016).

However, the possible influence of external climate forcing on these signatures could not be eliminated, and the causality behind the empirical relationships could not be identified. The dataset also

limited the authors to derive a definitive conclusion, as the oldest catchment lies in the driest region and the youngest catchment lies in a humid region (Yoshida and Troch, 2016). While the aridity index is a very useful and extensively used climate descriptor, it ignores any intra-annual variability of the climate signal. This intra-annual variability of climate can have a strong influence on the flow regime. Separating the effects of external climate and internal catchment properties is therefore

difficult if we are restricted to empirical analysis.

In this study, we investigate the robustness of the above-mentioned empirical relationship with the aid of process-based hydrological models by decoupling the effects of internal and external properties of the catchments. In addition, we examine the causality of such a simple relationship between hydrologic functioning and the catchment ages. More specifically, we address the following

questions: 1) what parameters and timescales of the hydrological model correlate with catchment age? 2) Do the correlated parameters and time scales corroborate the hypothesis? 3) Which of the catchment internal properties and external forcing climate has more contribution on the emergent empirical relationship?

## 2 Study materials and methods

### 2.1 Study catchments

We selected eight catchments (**Table 1**) out of 14 catchments studied by Yoshida and Troch (2016), all of which contain more than 50% volcanic rock coverage. We delineate the catchments using the digital elevation models (DEMs) of $10 \times 10$ m downloaded from the database (Geographic Survey Institute, Japan, 2015, online address: http://fgd.gesi.go.jp/download/). The stream networks were derived from digi-

tized GIS maps from the National Land Numerical Information website (Ministry of Land, Infrastructure and Transportation, Japan, 2015, online address: http://nlftp.mlit.go.jp/ksj/). The catchment ages were determined from the weighed average of the volcanic rock coverage and their ages, obtained from the Seamless Geological Map of Japan (Geological Survey of Japan, 2012). The age of the catchments ranges from 0.225 Ma to 82.2 Ma. Observed daily streamflow at each dam was obtained from the Japanese Dam

Database (online address: http://dam5.nilim.go.jp/dam/), all of which are not affected by human regulations as no major water use structures are located upstream.


**Table 1.** List of study catchments. $P$, $PE$ and $Q$ denotes annual precipitation, potential evaporation and streamflow, respectively. See Yoshida and Troch (2016) for more details.

| Catchment ID | Age (Ma) | Volcanic Coverage (-) | Area (km$^2$) | $P$ (mm/y) | $PE$ (mm/y) | $Q$ (mm/y) |
|---|---|---|---|---|---|---|
| SNK | 0.225 | 0.994 | 30.9 | 2454 | 932 | 1770 |
| ASE | 3.520 | 0.940 | 225.5 | 2215 | 903 | 1743 |
| KWM | 3.690 | 0.507 | 179.4 | 2333 | 947 | 1404 |
| SIM | 4.000 | 0.982 | 185.0 | 3112 | 1229 | 2267 |
| IKR | 10.700 | 0.653 | 271.2 | 1770 | 1006 | 1301 |
| AIM | 11.700 | 0.603 | 110.8 | 2652 | 1026 | 1679 |
| KAM | 12.800 | 0.621 | 195.3 | 2074 | 1024 | 1498 |
| HAZ | 82.200 | 0.623 | 217.0 | 1513 | 1249 | 573 |

## 2.2 Hydrological signatures

Hydrological signatures quantify the emergent properties of hydrological behaviors as index values, each of which reflects different temporal aspects of the catchment functions. Since different
temporal aspects of climate and catchment characteristics control different signatures, a sequential exploration of these signatures helps to decipher those controls better than direct comparison of complex hydrological models against observed streamflows.

The hydrological response of the study catchments are captured by two signatures in this study: the baseflow index (*BFI*) and the slope of the flow duration curve (*SFDC*). The baseflow index, defined
as the long-term average of baseflow fraction to the total streamflow, indicates the groundwater contribution to the catchment outflow (e.g., Vogel and Kroll, 1992; Kroll et al., 2004). The baseflow component was separated from the total streamflow with the following equation:

$$Q_b(t) = \epsilon Q_b(t-1) + \frac{1-\epsilon}{2}(Q(t) - Q(t-1)), \tag{1}$$

where $Q_b(t) = Q(t)$ when $Q_b(t) > Q(t)$; $Q(t)$ is total streamflow at time $t$, $Q_b(t)$ is the estimated
baseflow, and $\epsilon$ is a low-pass filter parameter (Arnold and Allen, 1999; Eckhardt, 2005). To maintain the consistency with other studies, the parameter $\epsilon$ was set at 0.925 for all catchments (Sawicz et al., 2011).

The slope in the middle part of the flow duration curve, calculated between the 33rd and 66th streamflow percentiles, was used as another index (Yadav et al., 2007; Zhang et al., 2008). This is
caluculated from

$$S_{fdc} = \frac{\log Q_{33} - \log Q_{66}}{0.66 - 0.33}, \tag{2}$$

where $Q_{66}$ and $Q_{33}$ are the streamflows that correspond to 66% and 33 % exceedance probability, respectively. High $S_{fdc}$ values indicate a variable flow regimes, while a low values suggest a more subdued responses.





### 2.3 hsB-SM model

The hillslope-storage Boussinesq soil moisture model (hsB-SM) is a physically-based and computationally efficient hydrological model that computes the energy and water balance of the land surface, the saturated subsurface flow and vertical exchange of water between root and saturated zones (Carrillo et al., 2011). Actual evapotranspiration values depend on the soil moisture conditions in the root zone, the root fraction and leaf area index (Teuling and Troch, 2005). Vertical fluxes between the root zone and the groundwater table (groundwater recharge and capillary rise) are calculated with the simplified unsaturated vertical flow formulations (Famiglietti and Wood, 1994). Three cases can occur: (1) capillary fringe below the root zone (2) capillary fringe in the root zone, and (3) saturation recharge, which is defined as the net flux of capillary uprise and gravitational drainage. A fraction of total recharge is assumed to contribute to a deep aquifer through fractured bedrock. Saturated subsurface flow is simulated with the hillslope-storage Boussinesq equation (Troch et al., 2003; Paniconi et al., 2003).

Climate data include precipitation, incoming short and long wave radiation, wind speed, temperature, and relative humidity. A 20 year meteorological dataset (from 1993 through 2012) was derived from the gridded meteorological data set of Yoshida and Troch (2016) and averaged for the entire catchment.

### 2.4 Calibration strategy

Certain model parameters can be assigend a priori based on landscape characteristics, while others are need to be identified through calibration procedures. We adopted a sequential calibration procedure to avoid parameter interactions and equifinality issues often encountered in automatic calibration. We first examined the sensitivity of parameters to hydrological signatures by perturbing one of 17 parameters at a time; and further investigated possible interactions between parameters by two-dimensional sensitivity analyses. Based on the sensitivity analysis, the parameters were categorized into three groups, which were linked to specific hydrological signatures, and the parameters in each group were independently calibrated with the Downhill Simplex Method (Nelder and Mead, 1965).

The master recession curve was compiled as the lower envelope of all the individual recession curves and used to calibrate the time constants for the deep and perched aquifers (Brutsaert and Nieber, 1977; Kirchner, 2009; Carrillo et al., 2011). The hsB-SM model conceptualized the early part of the master recession curve was composed of the deep and perched aquifers, while the late part was composed of the deep aquifer alone (Carrillo et al., 2011). By matching the simulated recession flows for the deep aquifers, we could isolate the deep aquifer contributions to the early part of the master recession curve. Once the deep aquifer contributions were isolated, the early part of the master recession curve was used for calibrating the hillslope-storage Boussinesq model (i.e., horizontal hydraulic





conductivity and drainable porosity). Then, the annual runoff coefficients and monthly regime curves
were used to calibrate model parameters related with the partitioning of water and energy at the land
surface (i.e., the depth of the root and transmission zones, respectively, and hydraulic conductivities
for the root and transmission zones). Finally, the parameters that control water flux in the unsaturated
zone were assigned from the hydrological signatures that describe daily streamflow fluctuations. We

tested both the slope of the flow duration curve (*SFDC*) and the baseflow index (*BFI*) as a descriptor
for calibrating the parameters in the unsaturated zone.

### 2.5    Decoupling the effects of internal properties and external forcing on hydrological functioning

Once we had a set of independently calibrated catchment models that could reproduce the empirical

relationship between hydrological signatures and catchment age, we used the models to separate the
roles of internal properties and external forcing on the emergent catchment behavior. Each parameterized model was exposed to each of the climates of the other catchments. We then calculated the
residuals of the simulated hydrologic signatures from the empirical regression lines presented by
Yoshida and Troch (2016), and evaluated the means of the residuals for each catchment and each

climate.

     If catchment internal properties are the only control on catchment behavior, the hydrological signatures obtained by a catchment model forced by other climates will not deviate from those obtained
with the local climate. The signatures should therefore plot along the horizontal line of the value
obtained when the climate was applied to the local catchment model; the residuals of the hydrologic

signatures from the regressed line per climate will be negative for younger catchments and positive
for older catchments (**Fig.1**, (a)). Also, the hydrological signatures obtained from one climate applied to the other catchment models will be the same as the signature value obtained by the climate
applied to the local model. This means that the mean residuals of the hydrologic signatures averaged
by catchment models should equal zero (**Fig.1**, (b)).

Conversely, if climate is the only control on the emergent hydrological signature, the resultant hydrological signatures of one catchment model forced by other climates would differ for each climate.
The residuals from the empirical regression line averaged by climate would be zero (**Fig.1**, (c)). Likewise, when one climate is used as the external driver of other models, the signatures should plot at
the same point as the original model. That is, the plot would shift horizontally on the age-signature

plane (**Fig.1**, (d)).

     The preceding explanation applies for the two end members, i.e., where either external forcing or
internal catchment properties are the only control on the emergent hydrologic signatures; in reality,
both sources can have an influence. If both exert influence, the signature values obtained by one
model forced with non-local climates should lie between the two extreme cases. Hence, the results

will plot between the horizontal line obtained by the model forced with local climate (the line shown





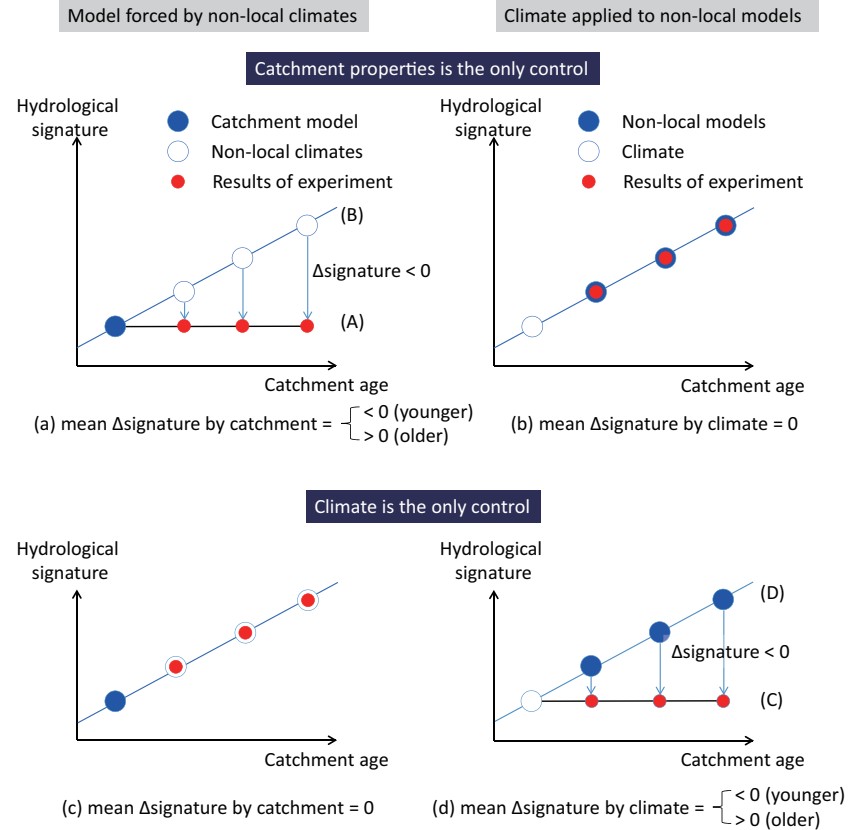

**Figure 1.** Schematic representation of numerical experiment if catchment internal property (above) and external climate (below) are the only control on hydrological signature, respectively: (a) and (c) model forced by other climates, (b) and (d) climate applied to other models.

as A in **Fig.1**) and the empirical regression line (line B in **Fig.1**). The deviation of the residuals of the hydrologic signatures from the empirical regression line indicates the level of influence of both controls on the emergent behaviors, i.e., the sensitivity of the signature to each of the controls or the range of the control. The plot would be closer to the line A in **Fig.1** if the catchment properties have

more control than the forcing climate, showing that the hydrological signature is more sensitive to the catchment properties, or that the range of catchment properties over which control is exerted is wider, generating greater deviation than do the forcing climate.

Overall, if both the internal properties and external forcing act as described above, the mean residuals averaged by catchment (mean Δsignature by catchment) should be negative for younger

catchments and positive for older catchments. As shown in **Fig.1** (b) and **Fig.1** (b), this relation is





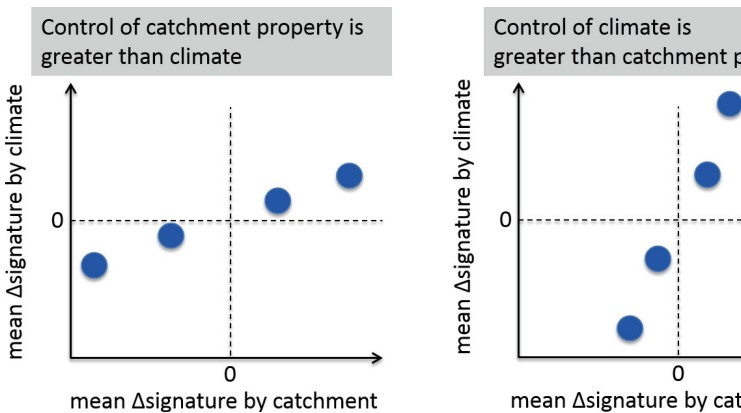

**Figure 2.** Plot of the mean resisudals of hydrological signatures averaged by catchment and by climate. The slope of the plot allows us to infer the relative levels of control exerted by the internal properties and external forcing in explaining the emergent hydrologic signatures: (a) greater controls of internal properties than those of climate, and (b) greater controls of climate than those of internal properties.

true for the mean residuals averaged by climate (mean $\Delta$signature by climate, **Fig.1**, (b)); hence, the plot of the mean $\Delta$signature by catchments versus those by climate would be positively correlated (**Fig.2**). The slope of the regression line would be zero (horizontal) if the internal catchment properties were the only control, and positive infinity (vertical) if the hydrological signatures were fully explained by only the external forcing (**Fig.2**). The slope of the plot therefore allows us to infer the relative levels of control exerted by the internal properties and external forcing in explaining the emergent hydrologic signatures.

## 3 Results

### 3.1 Comparison of the simulated and observed hydrological signatures

Comparing the simulated and observed annual runoff coefficients for each catchment shows that the calibrated models produced a satisfactory representation of the long-term water partitioning of the catchments (**Fig.3**). The seasonality of the water partitioning was also well represented in monthly regime curves, which confirmed the robustness of the model (not shown). The simulated *SFDC* values showed good agreement with the observed values, and the empirical linear regression of *SFDC* with catchment age was preserved (**Fig.4**). Although the runoff coefficients and *SFDC* were successfully represented by the calibrated models, the simulated *BFI* tended to be overestimated, especially for catchments that received a lot of winter snow, degrading the correlation between *BFI* and catchment age. This overestimation was likely due to the low-pass filter used for separating the baseflow from the total streamflow and the snowmelt. The simulated hydrograph during the snowmelt period



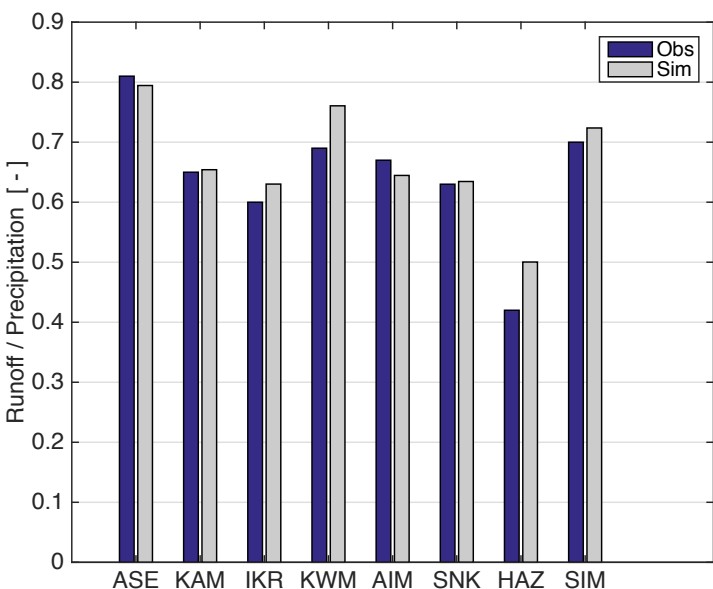

**Figure 3.** Observed and simulated runoff ratio for the study catchments.

was relatively smooth and most of the streamflow was regarded as baseflow, whereas the observed
       hydrographs displayed greater daily fluctuation such that a greater proportion of the total streamflow
       was identified as surface flow rather than baseflow. We therefore used *SFDC* as the key hydrological
       signature to represent the significant empirical correlation of hydrological response with catchment
       age.

225        The Nash–Sutcliffe Efficiency (*NSE*) was calculated for daily streamflow, *NSE* (Q), and for the
       log of daily streamflow, *NSE* (logQ).The *NSE* (logQ) values were superior to those of *NSE* (Q).
       The relatively poor representation of the daily streamflow is partly attributable to the low density
       of rain gauges in our study catchments, which required the precipitation data set to be corrected
       (Yoshida and Troch, 2016). The fact that the temporal resolution in our study (day) was longer than

the catchments concentration time also would have contributed to the relatively low *NSE* values,
       because we could not match the predicted and observed timing of the flood peaks (Carrillo et al.,
       2011). Although many of the *NSE* values are less than the threshold value of 0.5, values above
       which indicate that the simulation is behavioral (e.g., Santhi et al., 2001), we would argue that the
       models are acceptable because we calibrated them to represent the long-term catchment functioning,





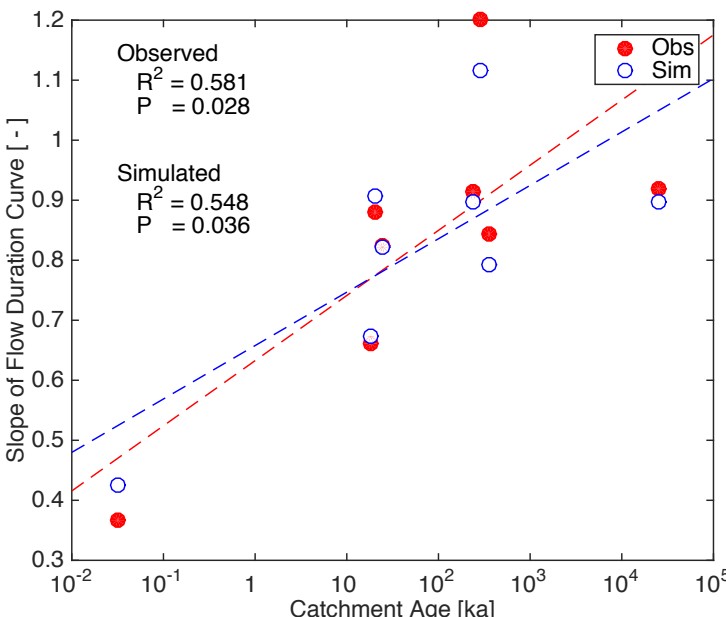

**Figure 4.** Observed and simulated slope of the flow duration curve (*SFDC*), showing *SFDC* is suitable descriptor of catchment signature for this study. The correlation of simulated *SFDC* with the catchment age is not deteriorated substantially when compared with the observed one.

which filters the input climate into the streamflow variance, rather than to accurately represent the timing of the daily peaks in the hydrographs.

### 3.2 Relationship between parameters and time scales with catchment age

The summary statistics of the linear regression between 17 model parameters of physical catchment properties and catchment age for the set of study catchments show that six of the parameters exhib-
ited significant correlation with catchment age (**Table 2**). The correlations with physical parameters related to the root zone were significant only because of the youngest catchment of the set; when that catchment was removed from the sample the correlation coefficients were very different (e.g., the root zone depth, $D_{rz}$). The correlation between the catchment age and the parameter for deep aquifer release $a$ (**Fig.5** (a)) suggested slower release of water from deep aquifers in younger catch-
ments, and the correlation was not deteriorated when the youngest catchment was removed. The parameters for perched aquifers, lateral hydraulic conductivity $K_h$, and drainable porosity $f$, were not correlated with catchment age. The transmission zone depth, $D_{tz}$, and the soil depth (sum of $D_{rz}$ and $D_{tz}$) exhibited significant decline with catchment bedrock age ($p = 0.018$).





**Table 2.** Linear regressions between the catchment age and the model parameters. Stars indicate significance ($p < 0.05$).

| Parameters | $p$ | $R^2$ | $R$ |
|---|---|---|---|
| Surface saturation conductivity (cm/d) | **0.009**\* | 0.706 | -0.84 |
| Saturation conductivity in root zone (cm/d) | 0.053 | 0.49 | -0.7 |
| Saturation conductivity in transition zone (cm/d) | **0.004**\* | 0.768 | -0.876 |
| Recharge fraction to deep aquifer (-) | 0.064 | 0.46 | -0.678 |
| Moisture content at wilting point (-) | 0.232 | 0.228 | 0.477 |
| Moisture content at critical point (-) | 0.058 | 0.477 | 0.691 |
| Vegetation height (m) | 0.089 | 0.407 | -0.638 |
| Root fraction (-) | 0.28 | 0.19 | -0.436 |
| Light use efficiency (-) | 0.278 | 0.191 | -0.437 |
| Root zone depth (m) | **0.029**\* | 0.577 | -0.759 |
| Transition zone depth (m) | **0.049**\* | 0.501 | -0.708 |
| Deep aquifer release a (-) | **0.015**\* | 0.653 | 0.808 |
| Drainable porosity (-) | 0.381 | 0.13 | -0.36 |
| Slope (-) | 0.478 | 0.087 | 0.295 |
| Lateral hydraulic conductivity (cm/d) | 0.419 | 0.112 | 0.334 |
| Deep aquifer release b (-) | 0.238 | 0.222 | -0.471 |
| Soil depth (m) | **0.018**\* | 0.632 | -0.795 |

Carrillo et al. (2011) defined the time scales that characterized the typical time period over which
water is stored, transmitted, and released in the components of the hsB-SM model. We examined the
linear regression between these time scales for the study catchments, including two dimensionless
numbers, and catchment age (**Table 3**). Five of the 10 time scales were significantly correlated with
age; however, as with the physical catchment parameters, the time scale parameters related to the
root zone were not as significant when the youngest catchment was removed from the plot. The
reservoir constant, which is the inverse of $a$, was significantly related to catchment age. The times
for filling, $T_{\mathrm{tf}}$, and emptying, $T_{\mathrm{te}}$, of the transmission zone declined with catchment age, showing it
took longer to fill or empty the transmission zones in younger catchments (**Fig.5** (b)).

The climate properties should be independent of catchment age; however, the aridity index and
the mean throughfall rate, which is a function of rainfall intensity and the leaf area index, were
linearly correlated with catchment age ($p = 0.041$). Thus, the significant correlations of simulated
signatures with catchment age need to be treated cautiously. We also explored the simulated fluxes
and found that the mean recharge rate to deep aquifers had a significant correlation ($p = 0.004$) with
catchment age (**Fig.5** (c)). The mean recharge rate to deep aquifers may be a function of several
factors related to the catchment internal properties and external climate, which caused us to consider
the interactions between internal properties and external forcing of the catchments.




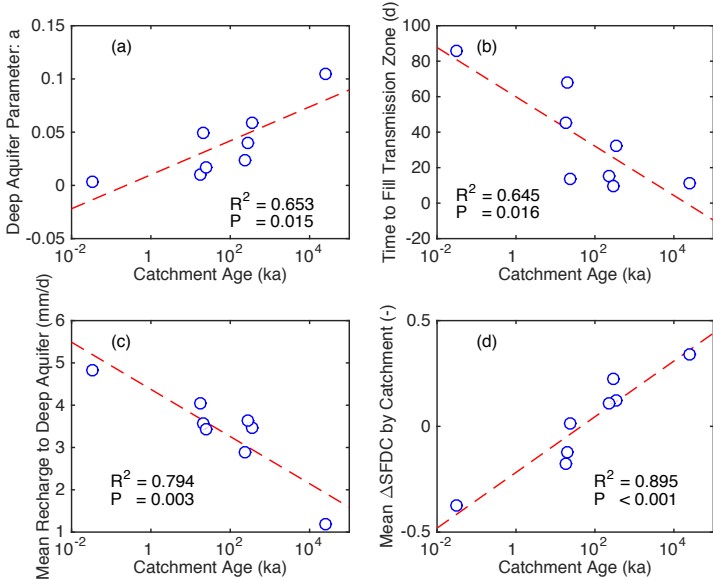

**Figure 5.** The significant correlation between the catchment age and (a) deep aquifer parameter $a$, (b) time to fill the transmission zone, (c) mean recharge rate to deep aquifer, and (d) mean $\Delta$SFDC by catchment.

**Table 3.** Linear regressions between the catchment age and the characteristics time scales and dimensionless numbers (see Carrillo et al., 2011 for details). Stars indicate significance ($p < 0.05$).

| Characteristics time scales and dimensionless numbers | $p$ | $R^2$ | $R$ |
|---|---|---|---|
| Filling root zone storage by rainfall (d) | 0.132 | 0.336 | -0.58 |
| Filling root zone storage by snow melt (d) | 0.134 | 0.333 | -0.577 |
| Emptying root zone storage by drainage (d) | **0.039**[*] | 0.535 | -0.731 |
| Emptying root zone storage by transpiration (d) | **0.028**[*] | 0.582 | -0.763 |
| Filling transmission storage (d) | **0.016**[*] | 0.645 | -0.803 |
| Emptying transmission storage (d) | **0.035**[*] | 0.549 | -0.741 |
| Advection-driven flow in perched aquifer (d) | 0.053 | 0.49 | -0.7 |
| Diffusion-driven flow in perched aquifer (d) | 0.704 | 0.026 | 0.16 |
| Peclet number (-) | 0.351 | 0.146 | 0.382 |
| Reservoir constant (-) | **0.004**[*] | 0.766 | -0.875 |





### 3.3 Decoupling the interactions between internal properties and external forcing

We used *SFDC* as the indicative hydrological signature to decipher the relative levels of control exerted by the internal properties and by the external forcing on emergent catchment behavior (**Fig.4**). Each parameterized model was forced by the non-local climates, and the residuals of the predicted 270 *SFDC* from the empirical regression line were averaged per catchment, hereafter referred to as the mean $\Delta SFDC$ by catchment. Similarly, the residuals of the *SFDC* were averaged per climate and defined as the mean $\Delta SFDC$ by climate.

We found a highly significant correlation between the mean $\Delta SFDC$ by catchment and mean $\Delta SFDC$ by climate, indicating that the external climate and internal catchment properties combined 275 in a consistent way to form the emergent behaviors of the catchments (**Fig.6**). The slope of 0.395 indicated that the catchment internal properties was a stronger control than the external climate.

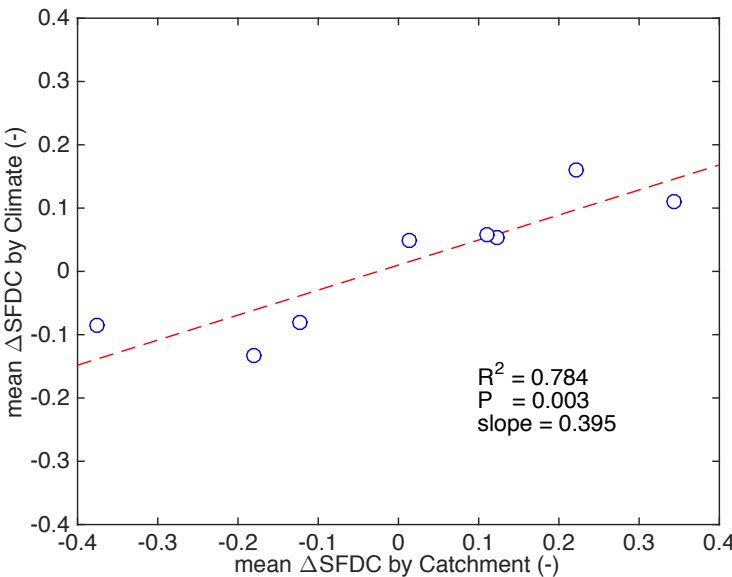

**Figure 6.** The plot of the mean $\Delta SFDC$ by catchment and by climate. The significant correlation suggests that the emergent catchment behavior is controlled by both internal catchment properties and external forcing of the catchments. The slope of the regression line, 0.395, may reflect the relative strength of each of the controls.

The mean $\Delta SFDC$ by catchment exhibited a significant positive correlation with catchment age ($p < 0.001$, **Fig.5** (d)). That is, the younger catchments tended to produce FDCs with gentle slopes independent of the climate, whereas the older ones tended to generate steeper slopes. This is stronger 280 evidence of the significant control of catchment age on the *SFDC* and corroborated the findings of Yoshida and Troch (2016) that the slope of the FDC increases as time progresses.





To further investigate the catchment internal properties, we examined the linear regression between the mean $\Delta SFDC$ by catchment and the physical parameters (Table S 2), the time scales (Table S 3), and the simulated fluxes of the hydrological model (Table S 4). The mean $\Delta SFDC$ by catchment was significantly correlated with four of the physical parameters (Table S 2). Like the correlations between $D_{\mathrm{rz}}$ and catchment age, the correlations with parameters related to the root zone storage were affected by the outlier youngest catchment. As a result, the plotted values for vertical hydraulic conductivity at the land surface, $K_{\mathrm{inf}}$, were clustered except for that of the youngest catchment, leaving the transmission zone depth $D_{\mathrm{tz}}$, deep aquifer release $a$, and the soil depth $D$ as the parameters that showed true significance. Also significantly correlated with the mean $\Delta SFDC$ by catchment were the time scales related to transmission zone filling, $T_{\mathrm{tf}}$, and emptying, $T_{\mathrm{te}}$, the reservoir constant (Table S 3),and the mean recharge rate to deep aquifers (Table S 4).

All of the physical parameters, time scale parameters, and fluxes that exhibited a significant correlation with the mean $\Delta SFDC$ by catchment were also correlated with the catchment age (Table S 2, S 3 and S 4), implying the strong influence of catchment age on the internal properties that influence the variability in partitioning of input water into streamflow. Strong significant correlations were found between the mean $\Delta SFDC$ by catchment and the time to fill the transmission zone, $T_{\mathrm{tf}}$ ($p = 0.004$), and the mean recharge rate to deep aquifers ($p = 0.011$).These results indicate that the transmission zone takes longer to fill in catchments that tend to produce more gently sloped FDCs; this is a function of both climate (e.g., rainfall intensity, evaporation rate) and catchment properties (e.g., soil depth, large storage in deep aquifers).

The only property that showed a significant correlation with the mean $\Delta SFDC$ by climate was the simulated mean transpiration rate ($p = 0.009$). The negative correlation suggested that higher transpiration rates were associated with steeper $SFDC$s.

## 4   Discussion

### 4.1   Hypothesis testing through numerical experiments

The motivation of this study was to test the robustness of the empirical relationship for catchment age vs hydrologic signatures postulated by Yoshida and Troch (2016) and to explore the causality of the relationship across a climate gradient. Within a numerical experiment, we replaced the empirical forcing climates and catchment properties with a calibrated process-based hydrological model, and revealed that the significant control exerted by the catchment internal properties on catchment tendency to generate a particular hydrological response was independent of the climate. When exposed to a non-local climate, all the catchments deviated from the empirical linear relationship between $SFDC$ and catchment age (Yoshida and Troch, 2016).

We evaluated the mean residuals of $SFDC$ averaged by catchment (mean $\Delta SFDC$ by catchment) and by climate (mean $\Delta SFDC$ by climate). The mean $\Delta SFDC$ by catchment, which indicates the





tendency of catchments to produce steep or gentle slopes in the FDC, exhibited significant negative correlations with catchment age. We investigated the causality of this relationship and found several time scale parameters and simulated fluxes to explain the relationship; that is, younger catchments, on average, require longer time for the transmission zone storages to fill and empty, have greater mean recharge rate to deep aquifers, and release water from deep aquifers more gradually than do older catchments. The strongest correlation with catchment age among these factors is the time to fill the transmission zone storage ($p = 0.004$), followed by the mean recharge rate to deep aquifers ($p = 0.011$) and the reservoir constant ($p = 0.014$). The partial least square regression, which can be used to identify latent variables among co-correlated predictors, confirmed that catchment age had the highest predictor loading on the response of the mean $\Delta SFDC$ by catchment. These results all corroborate the hypothesis of coevolution of volcanic catchments: younger catchments percolate more water vertically to the subsurface storage and have deep subsurface hydrologically active systems that release water slower than do older catchments, which have shallower systems that transfer water laterally rather than vertically (Jefferson et al., 2010; Yoshida and Troch, 2016). The consistency between the observational findings and the numerical experiments represents stronger evidence for the catchment coevolution theory than the results of the empirical study alone.

The parameterized transmission zone depth and total soil depth showed significant declines with catchment age ($p = 0.018$). This seems counterintuitive to findings for volcanic rocks in the Hawaiian Islands, where soils on younger rocks are shallow and coarse textured, whereas soils on older rock are deep and highly weathered (Lohse and Dietrich, 2005). A possible explanation for this contradiction is that the fractured bedrock and saprolite above young volcanic rock acts as a hydrologically active layer, whereas shallow clay layers in older catchments impede vertical flow, which thus result in effectively shallower soils.

Yoshida and Troch (2016) suggested that the model parameter that controls the recharge fraction to deep aquifers, $PTDA$, declines with catchment age. In the current study, the parameterized $PTDA$ exhibited a decreasing, though not quite significant ($p = 0.064$), trend with catchment age, which provides some support for this hypothesis.

### 4.2 Effects of external forcing on emergent hydrological signatures

The mean $\Delta SFDC$ by catchment exhibited a significant positive correlation with $\Delta SFDC$ by climate (**Fig.2**). The slope of the regression line suggests the relative influence of each factor in controlling the emergent hydrological behavior, varying from zero (100% catchment) to positive infinity (100% climate). The slope of 0.395 appears to suggest that the internal properties has stronger control than the external forcing (**Fig.2**). Despite the numerous significant correlations between catchment parameters and catchment age, the only parameter to show a significant correlation with mean $\Delta SFDC$ by climate, which is a function of the tendency of climate to cause $SFDC$ to deviate in a certain direction, was the mean transpiration rate ($p = 0.009$), leaving this as the sole candidate for a causative





role of climate on the hydrological signatures. This is an explicit and reasonable effect of climate
characteristics on hydrological behavior: the higher the transpiration rate, the less water is available
to support the middle part of the flow duration curves.

The significant relations found between climate characteristics and the catchment age suggest that
the external climate characteristics superimposed on the catchment internal properties enhanced the
relation between catchment age and hydrological responses. We found the mean recharge to deep
aquifers was significantly correlated with the aridity index and the mean throughfall rate; i.e., the
catchments with a more humid climate (lower aridity index) or higher throughfall rate contributed to
the increased recharge to deep aquifers. The mean recharge to deep aquifers was strongly correlated
with the catchment age ($p = 0.003$, **Fig.5**, (c)) by catchment ($p = 0.011$).and explained 68% of the
variability in the mean $\Delta SFDC$ The mean recharge to deep aquifers is likely to be a function of
both climate and catchment internal properties. The significant negative correlation of the mean
throughflow rate with catchment age in our data set may have partly explained the decline in the
mean recharge rate to deep aquifers. The predictor loading of the aridity index, $PTDA$ and the
mean throughfall rate estimated with the partial least square regression were similar and could not
identify any latent variables. The correlation of climate characteristics with catchment age might
blur the causal relationships, and hence the correlation of recharge to deep aquifers with catchment
age could be weakened when a different climate data set was used.

### 4.3 Towards improved predictions in ungauged basins

Recent studies suggested the importance of bedrock storage on hydrological behavior (e.g., Asano et al.,
2002; Sayama et al., 2011; Tague et al., 2008; Padilla et al., 2015). Studies have also indicated that
bedrock aquifers make a significant contribution to storm runoff generation, not only base flow
discharge (e.g., Mulholland, 1993; Freer et al., 2002; Padilla et al., 2015). Although these studies
suggest the significance of the storage volume and drainage characteristics of bedrock aquifers, the
variability in catchment characteristics has not yet been clarified. Estimating such characteristics
at large spatial scales and with an appropriate level of accuracy poses enormous problems because
hydrological characteristics of the bedrock aquifer is not just a soil property but reflects the perme-
ability, porosity, and degree of fracturing of bedrock.

Most studies on the regionalization of hydrological signatures have followed a statistical approach.
Catchment physical and climatic characteristics are usually considered to be as potential predictors
for the hydrological signatures, including catchment size, vegetation cover, and surface geology (e.g.,
Sawicz et al., 2011). Implicated factors have included soil classes and baseflow index (Croker et al.,
2003); available water capacity, soil depth, and soil texture classes (Mohamoud, 2008); HOST soil
classes (Holmes et al., 2002); percentage of volcanic/carbonate substrates (Rianna et al., 2011); and
spatial proximity (Sawicz et al., 2011). Despite these important empirical contributions, the literature
on process-based approaches is still sparse.



From a process perspective, the shape of the FDC (especially the middle section of the curve) can be influenced by the catchment's storage capacity (both surface and groundwater) and associated residence times (Lane and Lei, 1950), and how they interact with the seasonality of precipitation and potential evaporation. We would argue that the limitations of empirical studies can be complemented by numerical experiments in which systems are tested using catchment models (Carrillo et al., 2011; Troch et al., 2013). This study revealed that some of the catchment internal properties, especially those related to the fraction of recharge to bedrock and drainage properties of deep aquifers, are strongly correlated with catchment age. The significant correlation between model parameters and catchment age will help constrain the selection of process-based model parameters in ungauged catchments.

## 5 Conclusions

This study revealed that catchment internal properties, especially those related to recharge to bedrock and flux rates in deep aquifers, are strongly correlated with catchment (bedrock) age in volcanic catchments. The significant correlations between certain model parameters and catchment age will help constrain the process-based model parameters in ungauged catchments. The analysis also revealed that the external climate characteristics interact with the catchment internal properties in forming the catchment hydrological responses. The mean throughfall rate significantly declined with catchment age in our data set, and thus it enhanced the decline in recharge to deep aquifers with age. An increase in the mean transpiration rate was the only parameter that linked climate to a role in influencing hydrological signatures.

The strongest additional evidence for catchment coevolution that this study revealed was that the catchment age controls the slope of the flow duration curve ($SFDC$) independent of the forcing climate. The causes of this relationship appeared to be several simulated time-scale and flux parameters in the model. Younger catchments on average (1) required more time for the transmission zone storage to fill and empty and for water from deep aquifers to be released, and (2) had greater recharge to deep aquifers. These findings corroborate the coevolution hypothesis of volcanic catchments, which posits that younger catchments allow more water to percolate to the subsurface storage and have deeper hydrologically active systems that release water at a slower rate.



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
