# Peer review of "A process-based diagnosis of catchment coevolution in volcanic landscapes: synthesis of Newtonian and Darwinian approaches"

_Hydrology and Earth System Sciences, 2016_

## Referee Comment (RC1) · Anonymous Referee #1 · 24 Jul 2016

This study looks at the relations between topography and streamflow characteristics. It does so by using the interesting approach to switch catchments parameterisations and input time series. As much as I like the approach there are a number of concerns with the current version of the study:

There is a fundamental assumption that the input time series do not influence the parameter values. While one might hope for this (the parameters should represent the physical characteristics after all), many studies have shown, that parameter values actually are related to variables such as mean annual precip. This issue needs to be

addressed as it could largely influence the conclusions from this study.

P3L86: it is unclear why 8 out of 14 catchments were selected; this may sound a bit like cherry-picking. Please explain why/how only 8 catchments were selected here.

P5L138: The consideration of parameter uncertainty is not convincing. The sequential calibration is highly sensitive to the order of the parameter in calibration, and this order is not clearly described/motivated. Of course, this approach apparently reduces parameter interactions/uncertainty, but it does so by only investigating part of the parameter space. Just because one is not looking everywhere, does not mean parameter uncertainty is really reduced! I would recommend to consider parameter uncertainty explicitly by allowing for different parameter sets using some type of Monte Carlo approach (resulting in ranges of simulated streamflow characteristics).

Much of the analysis is based on the assumption that the model realistically can reproduce the observed runoff. There are two issues with this. The performance of the model in terms of NSE is not fully demonstrated, it is only mentioned that the NSE values were smaller 0.5 in many catchments (P9L232). This sounds rather alarming to me! Furthermore, even higher NSE values would not ensure that the different streamflow characteristics (signatures) would be realistically reproduced.

It also remains unclear how the different goodness-of-fit measures were combined (weighted mean). As the results largely depend on the parameterization and model performance, all these above issues are crucial for this study and need to be better addressed/described.

I am a bit confused by the term coevolution. This sounds fancy, but does the manuscript really deal with coevolution? I don't think so. Even if there is a relationship between topography and flow characteristics this does show necessarily any coevolution. Please clarify this term in the context of the manuscript (or omit it). The second part of the title (Newtonian/Darwinian) remains a total mystery to me, please explain what is meant here.

---

## Author Comment (AC1) · 3 Aug 2016

[Comment] This study looks at the relations between topography and streamflow characteristics. It does so by using the interesting approach to switch catchments parameterizations and input time series. As much as I like the approach there are a number of concerns with the current version of the study:

[Response] Thank you very much for taking the time to review our manuscript and for providing insightful comments. All of the concerns you raised have been addressed below. First, it is important to note that this study look at the relations between streamflow

characteristics and the age of the catchment, not with topography. We set out to infer the physical processes governing the catchment evolution (as it ages) with the aid of a process-based hydrological model. The model parameters are identified to reproduce the behavior of sub-models properly, rather than generating better objective functions (high NSE for instance). We examined several hydrological signatures including the annual and seasonal water partitioning at the land surface (Fig.3) and the slope of the flow duration curves (Fig.4). The parameter uncertainty is thus out of the scope of this paper.

[Comment] There is a fundamental assumption that the input time series do not influence the parameter values. While one might hope for this (the parameters should represent the physical characteristics after all), many studies have shown, that parameter values actually are related to variables such as mean annual precip. This issue needs to be addressed as it could largely influence the conclusions from this study.

[Response] We agree with you that the input time series will affect the parameter values, but since we used all available data (20 years) there is no other time period that we could use to see how sensitive the parameters are to the time series selected. Moreover, we don't know how this will affect the results, since we swap climates and models, and we look at how robust the predictions of certain signatures are to internal and external characteristics.

[Comment] P3L86: it is unclear why 8 out of 14 catchments were selected; this may sound a bit like cherry-picking. Please explain why/how only 8 catchments were selected here.

[Response] We selected eight catchments so that the age and climate gradients are maintained as the original samples. First, we plotted the age and aridity index of the study catchments of our previous study (Yoshida and Troch, 2016; Fig.1 in this response manuscript) and selected 4 clusters of catchments, i.e., the youngest (∼0.2 Ma), younger (∼3 Ma), older (∼10 Ma) and the oldest (∼80 Ma). We then selected

three catchments each from the 'younger' and 'older' clusters, each of which have different climate properties, and one catchment each from the 'youngest' and 'oldest'. Previous study of Carrillo et al. (2011) suggested the poor representation of hsB-SM in snow-dominant catchments; we thus selected both ASE and SIM catchments that are almost identical in terms of the age and aridity index, but are different as one of them is snow-dominant (ASE) while other is not (SIM). We will add text to the revised manuscript to make our selection procedure more transparent.

[Comment] P5L138: The consideration of parameter uncertainty is not convincing. The sequential calibration is highly sensitive to the order of the parameter in calibration, and this order is not clearly described/motivated. Of course, this approach apparently reduces parameter interactions/uncertainty, but it does so by only investigating part of the parameter space. Just because one is not looking everywhere, does not mean parameter uncertainty is really reduced! I would recommend to consider parameter uncertainty explicitly by allowing for different parameter sets using some type of Monte Carlo approach (resulting in ranges of simulated streamflow characteristics).

[Response] First, we used the same approach as in Carrillo et al (2011). The reason for this way of estimating the parameter values is that we want to make sure that each subcomponent of the model (shallow subsurface flow, deep groundwater, root zone dynamics etc) is behaving properly, we are not after a globally optimal model that may have the objective function right (high NSE for instance) but does a lousy job in getting the response of the subcomponents right. Monte Carlo simulations are thus out of the scope of this paper.

[Comment] Much of the analysis is based on the assumption that the model realistically can reproduce the observed runoff. There are two issues with this. The performance of the model in terms of NSE is not fully demonstrated, it is only mentioned that the NSE values were smaller 0.5 in many catchments (P9L232). This sounds rather alarming to me! Furthermore, even higher NSE values would not ensure that the different streamflow characteristics (signatures) would be realistically reproduced. It also remains unclear how the different goodness-of-fit measures were combined (weighted mean). As the results largely depend on the parameterization and model performance, all these above issues are crucial for this study and need to be better addressed/described.

[Response] The signature under investigation is the slope of the FDC, because this is the hydrologic signature that evolves with the age of the catchments. Figure 4 illustrates that the models do a good job in simulating this signature.

[Comment] I am a bit confused by the term coevolution. This sounds fancy, but does the manuscript really deal with coevolution? I don't think so. Even if there is a relationship between topography and flow characteristics this does show necessarily any coevolution. Please clarify this term in the context of the manuscript (or omit it). The second part of the title (Newtonian/Darwinian) remains a total mystery to me, please explain what is meant here.

[Response] Again, the paper is about the relation between the age of the catchments and flow characteristics, not about topography. The paper is about coevolution because we ask the question why does the slope of the flow duration curve changes predictably with age, and can we discover what catchment characteristics are responsible for this evolution? We build on the work of Jefferson et al. (2010) and Yoshida and Troch (2016) where this relationship was observed empirically, here we want to understand the mechanisms behind this, by testing the hypothesis put forward previously that with age the soils develop a shallow semi-impermeable layer that leads to faster flow paths and steeper slopes of the FDC. The Newtonian part refers to the mechanistic physics-based model that we use, the Darwinian part refers to the inter-comparison of catchments and the postulation of a hypothesis (and sequentially testing of this hypothesis) about what explains the difference in response and how it is related to catchment age (see Harman and Troch, 2014).

[Reference] Carrillo, G., Troch, P. A., Sivapalan, M., Wagener, T., Harman, C., and Sawicz, K.: Catchment classification: hydrological analysis of catchment behavior through process-based modeling along a climate gradient, Hydrology and Earth System Sciences, 15, 3411–3430, doi:10.5194/hess-15-3411-2011, http://dx.doi.org/10.5194/hess-15-3411-2011, 2011. Harman, C. and Troch, P. A.: What makes Darwinian hydrology" Darwinian"? Asking a different kind of question about landscapes, Hydrology and Earth System Sciences, 18, 417–433, 2014. Jefferson, A., Grant, G., Lewis, S., and Lancaster, S.: Coevolution of hydrology and topography on a basalt landscape in the Oregon Cascade Range, USA, Earth Surface Processes and Landforms, 35, 803–816, 2010.

Aridity Index vs Catchment Age (ka) plot with legend:
- ✳ Study catchments
- ● SNK (red)
- ● ASE (blue)
- ● ISB (black)
- ● IKR (green)
- ● AIM (cyan)
- ● KWM (magenta)
- ○ HZK (non-snow) (red outline)
- ○ SIM (non-snow) (blue outline)

**Fig. 1.** Selected eight catchments (circles) from the study catchments (stars indicate non-selected catchments) of Yoshida and Troch (2016)

---

## Referee Comment (RC2) · Anonymous Referee #2 · 10 Aug 2016

This paper is overall a nice model-based addition to the growing literature on volcanic catchment co-evolution. I really like how the authors have examined the effects of catchment internal properties versus climate on hydrologic response in way that can't be done in the real world. The results conform to the emerging conceptual view of how the hydrology of volcanic catchments changes as they age. However, I am concerned about over-interpretation of the results (because they fit so nicely!) relative to what can be said with confidence based on this modeling exercise.

The first referee has provided some really important points about the parameter un-

certainty and estimation and fairly low model performance as measured by the NSE. While I respect the authors' point that they are able to reasonably replicate the observed slopes of the flow duration curves (SFDC), as shown in the figure, the overall lack of fit of the model to the data raises questions about how much credence we should give to model results for the simulated combinations of climate and catchment characteristics. How much of the SFDC residual is due to model performance versus the processes actually under study? It seems like a more complete uncertainty analysis that attempts to propagate the model uncertainty through the results would be quite useful in interpreting the findings.

Beyond these major points, I have a few secondary points.

The authors have assessed how a large number of catchment parameters change with catchment age and climate. They use a cut of $p<0.05$ as a test of statistical significance of these regressions, without making a correction for multiple comparisons that can lead to false positives. I suggest that the authors apply the standard Bonferroni correction and adjust the p-value for significance accordingly.

In the paragraph around line 65, the authors make a statement about the changes over time in clay layers, chemical weathering, vertical recharge and shallow subsurface flow in volcanic catchments. They cite Jefferson et al (2010) and their previous paper. It should be noted that neither of these papers actually directly observed those processes. Instead both papers were empirical studies of change in stream hydrographs with catchment age that put forward these ideas as possible explanations for the hydrologic signatures. There is literature on clays and chemical weathering of basalts with respect to soil development (c.f., work by Oliver Chadwick and Peter Vitousek), and the work of Lohse and Dietrich (2005) adds some hydrological context to the soil development story.

Around line 72, the mention of the aridity index doesn't make sense for readers unfamiliar with the previous paper. Perhaps the authors should provide more context.

[Figure]

The conceptualization in Figure 1 is very nice, but mean delta-signature should be defined within the figure or its caption. Also, is the only way to know what the slope of B looks like through an empirical set of watersheds of different ages? How does this limit the utility of the framework you put forward?

In Figures 4 and 5, the x-axis is labeled with units of ka. I believe it should be Ma, based on the assembled catchment ages.

---

## Author Comment (AC2) · 2 Sep 2016

[Comment]

This paper is overall a nice model-based addition to the growing literature on volcanic catchment co-evolution. I really like how the authors have examined the effects of catchment internal properties versus climate on hydrologic response in way that can't be done in the real world. The results conform to the emerging conceptual view of how the hydrology of volcanic catchments changes as they age. However, I am concerned about over-interpretation of the results (because they fit so nicely!) relative to what can

be said with confidence based on this modeling exercise.

The first referee has provided some really important points about the parameter uncertainty and estimation and fairly low model performance as measured by the NSE. While I respect the authors' point that they are able to reasonably replicate the observed slopes of the flow duration curves (SFDC), as shown in the figure, the overall lack of fit of the model to the data raises questions about how much credence we should give to model results for the simulated combinations of climate and catchment characteristics. How much of the SFDC residual is due to model performance versus the processes actually under study? It seems like a more complete uncertainty analysis that attempts to propagate the model uncertainty through the results would be quite useful in interpreting the findings.

[Response]

We compiled figures that present the various aspects of model performance (flow duration curve for entire period, one-year cumulative streamflow curve and one-year daily hydrograph) with the values of NSE (Q), NSE (logQ), NSE(FDC) and NSE(CF) in the caption (Figures 1-8). All models successfully captured the flow duration curves (top left in each figure) and cumulative flow (top right), but not necessarily good with respect to the daily streamflow (bottom). The NSE values are low partly because of the timing of the simulated flood peaks do not coincide with the observed ones. The timing of flood peaks can be delayed with the flood concentration time, which represent catchments' hydraulic characteristics at the land surface (Carrillo et al., 2011); however, we could not apply the delay function in this study because the flood concentration time is less than the simulation interval of one day.

Overall, we would argue that even though the NSE values with daily streamflow are low, the frequency of flows is captured well. So, the model does poorly on timing the flow but performs well on generating the 'right amount of flow'. While we agree that the model parameters still have uncertainty, they are identified as we intended and the

model errors are negligible.

[Comment]

Beyond these major points, I have a few secondary points. The authors have assessed how a large number of catchment parameters change with catchment age and climate. They use a cut of $p < 0.05$ as a test of statistical significance of these regressions, without making a correction for multiple comparisons that can lead to false positives. I suggest that the authors apply the standard Bonferroni correction and adjust the p-value for significance accordingly.

[Response]

The reviewer is correct that we tested two hypotheses simultaneously; 1) presence/absence of coevolution (with respect to the catchment age), and 2) degree of catchment/climate controls on flow characteristics (with respect to $\Delta$SFDC by catchment/climate). In Bonferroni correction, the threshold for the statistical significance (i.e., p-value of 0.05) should be divided by the number of hypotheses. With the p-value of 0.025 (= 0.05/2), some of the relations we suggested as significant in the previous version of manuscript should be perceived as not significant (or significant not because of the coevolution only but climate might have affected); those are the relation between catchment age and timescales for emptying root zone by drainage ($p=0.039$) and by transpiration ($p=0.028$), for emptying transmission storage ($p=0.035$), the aridity index ($p=0.033$), and the mean throughfall rate ($p=0.041$). While these relations should be interpreted as not significant, the main conclusion suggested in Figure 5 and 6 would not be altered because their p-values were less than 0.025.

We would add this correction and the consequence in the revised manuscript.

[Comment]

In the paragraph around line 65, the authors make a statement about the changes over time in clay layers, chemical weathering, vertical recharge and shallow subsurface flow

in volcanic catchments. They cite Jefferson et al (2010) and their previous paper. It should be noted that neither of these papers actually directly observed those processes. Instead both papers were empirical studies of change in stream hydrographs with catchment age that put forward these ideas as possible explanations for the hydrologic signatures. There is literature on clays and chemical weathering of basalts with respect to soil development (c.f., work by Oliver Chadwick and Peter Vitousek), and the work of Lohse and Dietrich (2005) adds some hydrological context to the soil development story.

[Response]

The references for this part will be changed in the revised manuscript to the papers on soil development in volcanic landscapes as you suggested.

[Comment]

Around line 72, the mention of the aridity index doesn't make sense for readers unfamiliar with the previous paper. Perhaps the authors should provide more context.

[Response]

The description of the aridity index will be added to the revised manuscript.

[Comment]

The conceptualization in Figure 1 is very nice, but mean delta-signature should be defined within the figure or its caption. Also, is the only way to know what the slope of B looks like through an empirical set of watersheds of different ages? How does this limit the utility of the framework you put forward?

[Response]

The 'delta-signature' refers to the deviation of hydrological signature from the empirical line shown as line (B) or (D) in Figure 1. The 'delta-signature' in this study is $\Delta$SFDC as described later; however, we did not use the term, $\Delta$SFDC, here to make this conceptual figure as a general framework of this study. Yes, the only way to know the slope of B is based on the empirical studies at this point.

[Comment]

In Figures 4 and 5, the x-axis is labeled with units of ka. I believe it should be Ma, based on the assembled catchment ages.

[Response]

Thank you for pointing this out.

———————————————————

**Flow duration curve**

**Cumulative water balance**

**Hydrograph**

**Fig. 1.** Combined figures of flow duration curve (top left), cumulative flow (top right) and one-year hydrograph (bottom) in AIM (NSE(Q) = 0.360; NSE(logQ) = 0.633; NSE(FDC) = 0.964; NSE(CF) = 0.995).

**Flow duration curve**

**Cumulative water balance**

**Hydrograph**

**Fig. 2.** Combined figures of flow duration curve (top left), cumulative flow (top right) and one-year hydrograph (bottom) in ASE (NSE(Q) = 0.419; NSE(logQ) = 0.689; NSE(FDC) = 0.794; NSE(CF) = 0.994).

**Flow duration curve**

**Cumulative water balance**

**Hydrograph**

**Fig. 3.** Combined figures of flow duration curve (top left), cumulative flow (top right) and one-year hydrograph (bottom) in HAZ (NSE(Q) = 0.209; NSE(logQ) = 0.430; NSE(FDC) = 0.894; NSE(CF) = 0.965).

**Fig. 4.** Combined figures of flow duration curve (top left), cumulative flow (top right) and one-year hydrograph (bottom) in IKR (NSE(Q) = 0.428; NSE(logQ) = 0.510; NSE(FDC) = 0.968; NSE(CF) = 0.985).

**Flow duration curve**

**Cumulative water balance**

**Hydrograph**

**Fig. 5.** Combined figures of flow duration curve (top left), cumulative flow (top right) and one-year hydrograph (bottom) in KAM (NSE(Q) = 0.696; NSE(logQ) = 0.422; NSE(FDC) = 0.941; NSE(CF) = 0.934).

**Fig. 6.** Combined figures of flow duration curve (top left), cumulative flow (top right) and one-year hydrograph (bottom) in KWM (NSE(Q) = 0.302; NSE(logQ) = 0.613; NSE(FDC) = 0.883; NSE(CF) = 0.916).

**Fig. 7.** Combined figures of flow duration curve (top left), cumulative flow (top right) and one-year hydrograph (bottom) in SIM (NSE(Q) = 0.711; NSE(logQ) = 0.647; NSE(FDC) = 0.937; NSE(CF) = 0.996).

[Figure]

**Fig. 8.** Combined figures of flow duration curve (top left), cumulative flow (top right) and one-year hydrograph (bottom) in SNK (NSE(Q) = 0.358; NSE(logQ) = 0.365; NSE(FDC) = 0.947; NSE(CF) = 0.941).